# Joint Geometrical and Statistical Domain Adaptation for Cross-domain Code Vulnerability Detection

**Qianjin Du[1], Shiji Zhou[2], Xiaohui Kuang[3], Gang Zhao[3*], Jidong Zhai[1*]**

[1]Department of Computer Science and Technology, Tsinghua University
[2]Department of Automation, Tsinghua University
[3]National Key Laboratory of Science and Technology on Information System Security
dqj20@mails.tsinghua.edu.cn; zhaijidong@tsinghua.edu.cn
zhoushiji00@gmail.com
xhkuang@bupt.edu.cn; bisezhaog@163.com

## Abstract

In code vulnerability detection tasks, a detector trained on a label-rich source domain fails to provide accurate prediction on new or unseen target domains due to the lack of labeled training data on target domains. Previous studies mainly utilize domain adaptation to perform cross-domain vulnerability detection. But they ignore the negative effect of private semantic characteristics of the target domain for domain alignment, which easily causes the problem of negative transfer. In addition, these methods forcibly reduce the distribution discrepancy between domains and do not take into account the interference of irrelevant target instances for distributional domain alignment, which leads to the problem of excessive alignment. To address the above issues, we propose a novel cross-domain code vulnerability detection framework named MN-CRI. Specifically, we introduce mutual nearest neighbor contrastive learning to align the source domain and target domain geometrically, which could align the common semantic characteristics of two domains and separate out the private semantic characteristics of each domain. Furthermore, we introduce an instance re-weighting scheme to alleviate the problem of excessive alignment. This scheme dynamically assign different weights to instances, reducing the contribution of irrelevant instances so as to achieve better domain alignment. Finally, extensive experiments demonstrate that MNCRI significantly outperforms state-of-the-art cross-domain code vulnerability detection methods by a large margin.

## 1 Introduction

With the development of the Internet, the number of vulnerabilities in software has increased rapidly, which leaves software to face huge security threats. There are multiple efforts (Lin et al., 2017; Grieco et al., 2016; Li et al., 2018; Zou et al., 2019; Zhou et al., 2019; Feng et al., 2020) have attempted to introduce deep learning and NLP techniques for automated vulnerability detection. Those deep learning-based detection methods mainly focus on in-domain code vulnerability detection. That is, the training and test datasets are assumed to be drawn from the same distribution *(i.e., from the same software project)*. However, in practice, most software projects lack labeled vulnerability datasets. Labeling data is labor-intensive and expensive, which requires that the models trained on a label-rich source domain (i.e., a software project with abundant labeled data) are applied to a new target domain (i.e., a software project without labeled data). Owing to program style, application scenarios, and other factors, different software projects may obey different probability distributions. The models trained on the source domain suffer from severe performance loss on the target domain due to the obvious distribution discrepancy between two domains. To solve this problem, some research works have attempted to introduce domain adaptation to perform cross-domain code vulnerability detection. Domain adaptation is a technique that aims to eliminate distribution discrepancy and transfer knowledge from a labeled source domain to an unlabeled target domain (Ganin and Lempitsky, 2015). Lin et al. (2018) first proposed a function-level cross-domain code vulnerability detection framework. But they only hypothesize that the learned high-level representations are transferable, and they do not propose a method to explicitly eliminate the domain discrepancy. Lin et al. (2019) proposed a cross-domain detection framework that utilizes two LSTMs to learn two groups of transferable feature representations, and then combine both groups of feature representations to train a classifier. Nguyen et al. (2019) proposed a new code domain adaptation framework named SCDAN that incorporated conventional domain adaptation techniques such as

---

*Corresponding authors.

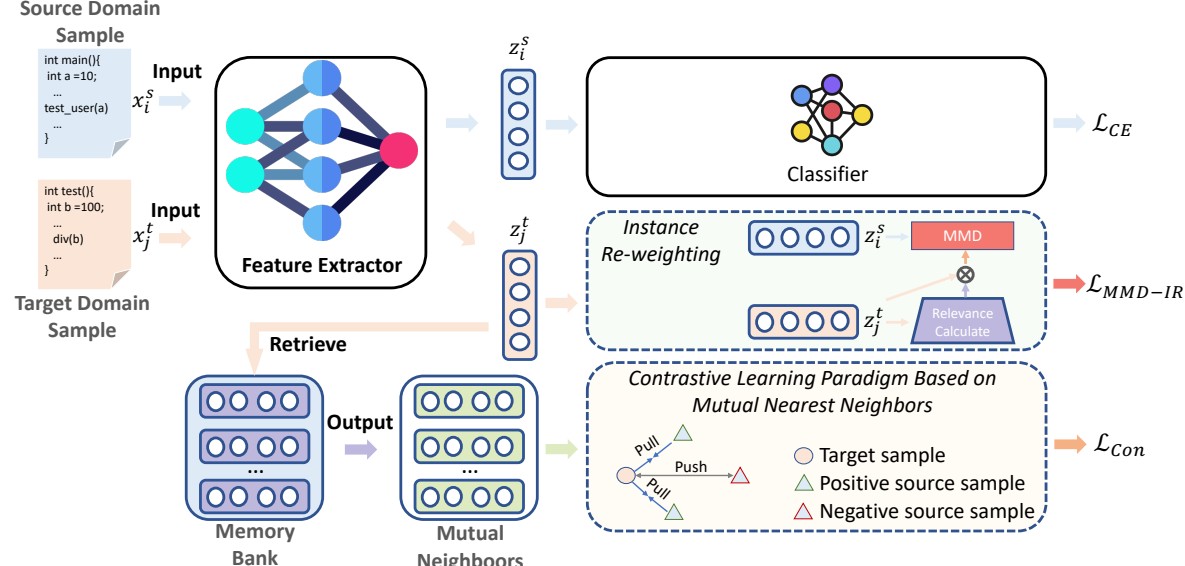

Figure 1: The framework of MNCRI.

DDAN, MMD, and DIRT-T (Ganin and Lempitsky, 2015; Long et al., 2015; Shu et al., 2018). CD-VulD (Liu et al., 2020) employed a metric transfer learning framework (MTLF) (Xu et al., 2017) to learn cross-domain representations by minimizing the distribution divergence between the source domain and the target domain. Lastly, Li et al. (2023) proposed a detection framework called VulGDA that utilized a graph embedding method to extract comprehensive vulnerability features and a feature generator to generate domain-invariant feature representations by minimizing Maximum Mean Discrepancy (MMD).

Although the above-mentioned research works achieve considerable performance improvements, there still exist two problems that restrict further performance improvements: (1) In practice, different domains (i.e., different software projects) share common semantic characteristics and while having their own private semantic characteristics due to different application scenarios, coding style, and code implementation, etc. Previous studies reduce the domain discrepancy by forcibly minimizing divergence of the entire source domain and the target domain. They ignore semantic differences between two domains and allow samples with private semantic characteristics to participate in domain alignment, which is prone to the problem of negative transfer. Eliminating the domain discrepancy should be constrained on only aligning the common semantic characteristics (also known as the common knowledge) between two domains

and the respective private semantic characteristics should be separated simultaneously. (2) There will always exist some target instances that are not relevant to source instances. Current methods treat the instances equally when conducting statistical distribution domain alignment. Intuitively, the relevant instances should be given larger weights, and irrelevant instances should be assigned smaller weights so as to decrease their importance and contribution to domain alignment. Ignoring the relevance of instances will make models learn sub-optimal cross-domain representations. To address these issues, we propose a novel cross-domain code vulnerability detection framework named MNCRI (as illustrated in Figure 1) based on mutual nearest neighbor contrastive learning and the instance re-weighting scheme, which jointly achieves geometric domain alignment and statistical domain alignment. First, we argue that geometric domain alignment should follow a principle: semantically consistent instances should be geometrically adjacent to each other, whether within or across domains. Based on this principle, we introduce mutual nearest neighbor contrastive learning to achieve the alignment of common semantic characteristics. Specifically, we construct mutual nearest neighbor pairs across the source domain and the target domain by leveraging the mutual neighbor rule. Then we introduce a contrastive loss function that aims at reducing the feature discrepancy of mutual nearest neighbor pairs. By minimizing this contrastive loss, we achieve the alignment of

common semantic characteristics across domains and separate out private semantic characteristics of each domain. Second, to alleviate the problem of excessive alignment, we introduce the instance re-weighting scheme that assigns different weights to different target domain instances based on the relevance with source domain instances. In this way, the effect of some target domain instances that are irrelevant to source domain instances is reduced and the effect of some relevant target domain instances is highlighted, which is beneficial for reducing the distribution discrepancy. Finally, we conduct extensive experiments to verify the effectiveness of our proposed method. The experimental results show that the proposed method outperforms current baseline methods by a large margin.

In summary, our contributions are as follows:

- We propose a novel cross-domain code vulnerability detection framework named MNCRI, which reduces the domain discrepancy from two perspectives of geometries and statistical distributions by exploiting the mutual nearest neighbor contrastive learning and the instance re-weighting scheme.

- We introduce mutual nearest neighbors contrastive learning to solve the problem of negative transfer, which explores instances relationships between domains and achieves the domain alignment geometrically.

- We incorporate an instance re-weighting scheme into the statistical domain alignment, which dynamically weights different instances according to the relevance between instances to eliminate the negative effect of irrelevant instances for domain alignment.

- We conduct extensive experiments to verify the effectiveness of MNCRI. Experimental results show that MNCRI outperforms current cross-domain detection methods and obtain obvious performance improvements.

## 2 Related Work

### 2.1 Cross-domain Code Vulnerability Detection

Since different software projects have different code styles, application scenarios and other factors, they may obey different distributions. Therefore, when a code vulnerability detection model is applied to a new no-label project, its performance drop sharply. To address this issue, researchers have attempted to introduce domain-adaptation techniques into cross-domain code vulnerability detection tasks. Nam et al. (2013) utilized migration component analysis to map software metrics in the source and target domains to the same feature space. Hellendoorn et al. (2020) attempted to utilize local syntactic features and global semantic features to perform file-level cross-domain vulnerability detection. Lin et al. (2018) first proposed a cross-domain code vulnerability detection framework at the function-level, which extracts features from abstract syntax trees (ASTs) of functions and then feeds extracted features into a long short-term memory recurrent neural network (LSTM) to learn higher-level representations. But they only hypothesize that the learned high-level representations are transferable, and they do not propose an effective method to explicitly eliminate the discrepancy between the source domain and the target domain. Lin et al. (2019) presented a cross-domain detection framework that employed two LSTMs to learn two groups of transferable feature representations. These representations were then combined to train a classifier. Nguyen et al. (2019) proposed a novel code domain adaptation network called SCDAN. SCDAN incorporated conventional domain adaptation network techniques such as DDAN, MMD, and DIRT-T (Ganin and Lempitsky, 2015; Long et al., 2015; Shu et al., 2018) to eliminate distribution divergence. To further tackle the distribution divergence, CD-VulD (Liu et al., 2020) deployed the metric transfer learning framework (MTLF) (Xu et al., 2017) to learn domain-invariant representations by minimizing Mahalanobis distance. Li et al. (2023) proposed a cross-domain detection framework called VulGDA, which employed a graph embedding approach to extract comprehensive vulnerability features and a feature generator to generate domain-invariant feature representations by minimizing the domain discrepancy.

### 2.2 Contrastive Learning

Contrastive learning in the latent space has recently shown great promise, which aims to make the representation of a given anchor data to be similar to its positive pairs and dissimilar to its negative pairs (Tian et al., 2020; Chen et al., 2020; Khosla et al., 2020). Many state-of-the-art methods for represen-

tation learning tasks are based on the contrastive learning framework (Chen and He, 2021). Previous studies construct positive samples by augmenting original samples. In this paper, we utilize mutual nearest neighbors as positive pairs to achieve feature alignment between domains.

## 3 Methodology

As illustrated in Figure 1, the architecture of MN-CRI contains two main components: 1) the mutual nearest neighbor contrastive learning module, which performs contrastive learning based on mutual nearest neighbors for better geometric domain alignment; 2) the statistical distribution domain alignment module equipped with the instance reweighting scheme, which reduces the distribution discrepancy and achieves the domain alignment statistically. Below we elaborate the details of MN-CRI.

### 3.1 Problem Formulation

We formalize vulnerability detection as a binary classification problem. In cross-domain vulnerability detection, given a labeled source domain set $D_s = \{(x_i^s, y_i^s)\}_{i=1}^{n_s}$ including $n_s$ instances $x_i^s$, it aims to predict the label $y_i^t$ of the target sample $x_i^t$ in an unlabeled target domain set $D_t = \{(x_i^t)\}_{i=1}^{n_t}$, where $y_i^s \in \{0, 1\}$ and $y_i^t \in \{0, 1\}$ (1 means the sample is vulnerable and 0 means the sample is non-vulnerable). $n_s$ and $n_t$ are the number of the source domain set and the target domain set, respectively. As illustrated in Figure 1, our model consists of two basic modules: (1) a feature extractor $f(\cdot)$ that maps the code sample $x_i$ into the embedding representation $z_i$ ($z_i = f(x_i)$). (2) a classifier $g(\cdot)$ to classify the class $y_i$ of the sample $x_i$ ($y_i = g(f(x_i))$).

### 3.2 Geometric Domain Alignment by Mutual Nearest Neighbor Contrastive Learning Paradigm

In supervised learning scenarios, the cross-entropy loss on the source domain is defined as:

$$
\begin{aligned}
\mathcal{L}_{CE} = - \frac{1}{n_s} \sum_{i=1}^{n_s} & y_i^s \log(g(f(x_i^s)))+ \\
& (1 - y_i^s) \log(1 - g(f(x_i^s)))
\end{aligned}
\tag{1}
$$

Due to the domain discrepancy between the source domain and the target domain, the performance drops sharply when transfer the model trained on

source domain to the target domain. Previous methods mainly focus on minimizing the global distribution divergence between domains to achieve the domain alignment, such as Maximum Mean Discrepancy (MMD) (Yan et al., 2017; Li et al., 2023). In practice, the target domain may preserve its own private semantic characteristics that do not appear in the source domain. Previous methods do not consider domain private semantic characteristics and forcefully align the global distributions of two domains, which may cause the problem of negative transfer. In manifold learning (McInnes et al., 2018; Chen et al., 2022), the concept of geometric nearest neighbors is commonly used to describe the similarities of semantic characteristics. For different domains, the geometrically nearest neighbors can be viewed as the most similar samples that contain the same semantic characteristics. Therefore, to align the common semantic characteristics between domains and separate out private semantic characteristics of each domain, we introduce mutual nearest neighbors contrastive learning.

Specifically, for a target domain sample $x_i^t$, we retrieve its corresponding $k$ nearest neighbors $\mathcal{NN}_i^s$ in the source domain. In the same way, for a source domain sample $x_j^s$, we can also obtain its corresponding $k$ nearest neighbors $\mathcal{NN}_j^t$ in the target domain. Then mutual nearest neighbors can be constructed by following the rule: if the source sample $x_j^s$ and the target sample $x_i^t$ are contained in each other's nearest neighbors, they are regarded as mutual nearest neighbors. After that, we build a relationship matrix to describe the mutual nearest neighbor relationship between source domain samples and target domain samples. Formally, the relationship matrix is defined as: $M^{st} \in \mathbb{R}^{n_s \times n_t}$ where $M_{i,j}^{st} = 1$ if only $x_i^t$ and $x_j^s$ is the mutual nearest neighbor pair; otherwise $M_{i,j}^{st} = 0$. Besides, to fully utilize supervised information of the source domain and learn better discriminative features for classification, we also build in-domain relationship matrix $M^{ss}$ according to ground-truth labels of the source domain.

In geometric domain alignment, we hope that built mutual nearest neighbor pairs can be closer to each other in the feature space, while non-geometrically samples should be far away from each other. In this paper, we introduce an contrastive learning loss to achieve this goal.

For each training sample $x_i$, its positive sample sets $P_i$ and negative sample sets $N_i$ in the memory

bank can be obtained by relationship matrices $M^{ss}$ and $M^{st}$. The contrastive loss is formulated by:

$$\mathcal{L}_{Con} = -\sum_{i=1}^{n_s+n_t} \sum_{j\in P_i} \log \varphi_{i,j}$$

$$\log \varphi_{i,j} = \frac{exp(z_i z_j/\tau)}{\sum_{p\in P_i} exp(z_i z_p/\tau) + \sum_{q\in N_i} exp(z_i z_q/\tau)}$$

(2)

where $\tau$ is a temperature parameter. By optimizing the contrastive loss, it makes the model learn geometric neighbor relationships between the source domain and the target domain and makes the feature representations of samples with common semantic characteristics closer in the feature space.

### 3.3 Statistical Domain Alignment by Instance Re-weighting Scheme

As depicted in Section 3.2, the mutual nearest neighbors contrastive learning could reduce the geometrical domain discrepancy and achieve better geometrical domain alignment. In practice, reducing the distributional divergence between domains and achieving the statistical domain alignment are also important for cross-domain code vulnerability detection. MMD (Gretton et al., 2008)) is a popular estimator to calculate the degree of statistical alignment of data distribution between two domains, and it is defined as follows:

$$\mathcal{L}_{MMD} = \| \frac{1}{U^2} \sum_{i=1}^{U} \sum_{j=1}^{U} k(z_i^s, z_j^s) - \frac{2}{UW} \sum_{i=1}^{U} \sum_{j=1}^{W} k(z_i^s, z_j^t)$$

$$+ \frac{1}{W^2} \sum_{i=1}^{W} \sum_{j=1}^{W} k(z_i^t, z_j^t)\|_{\mathcal{H}}$$

(3)

Previous studies (Liu et al., 2020; Li et al., 2023) perform statistical domain alignment by minimizing Maximum Mean Discrepancy (MMD). However, there will always exist some instances that are not relevant to the source instances. Previous methods forcefully reduce the global distributional divergence between domains and ignore the negative effect of irrelevant instances, which easily leads to the problem of excessive alignment. Ideally, relevant target instances should be assigned large weights and irrelevant instances should be assigned small weights during the domain alignment. Thus, in order to achieve this goal, we propose to introduce the instance re-weighting scheme into the statistical domain alignment. Specifically, we adopt the prediction confidence of the classifier trained on the source domain for target instances

as the weights of target instances during statistical domain alignment. This process can be formulated as:

$$\mathcal{L}_{MMD-IR} = \| \frac{1}{U^2} \sum_{i=1}^{U} \sum_{j=1}^{U} k(z_i^s, z_j^s)$$

$$-\frac{2}{UW} \sum_{i=1}^{U} \sum_{j=1}^{W} f(z_j^t) k(z_i^s, z_j^t) + \frac{1}{W^2} \sum_{i=1}^{W} \sum_{j=1}^{W} k(z_i^t, z_j^t)\|_{\mathcal{H}}$$

$$f(z_j^t) = 2 * |Prob(y_j^t = 1 | z_j^t) - 0.5|$$

(4)

where $Prob(y_j^t = 1|z_j^t)$ denotes the prediction probability that the classifier predicts the target instance $x_i^t$ is vulnerable. Introducing the instance re-weighting scheme degrade the impact of some target instances that are not relevant to source instances, which is helpful for reducing the distributional divergence between domains and achieving better statistical domain alignment.

### 3.4 Overall Objective

MNCRI is proposed to jointly minimize the cross entropy loss of the source domain instances and reduce the divergence between the source domain and target domain. The total objective function in MNCRI is formulated as:

$$\mathcal{L} = \mathcal{L}_{CE} + \lambda\mathcal{L}_{Con} + \mu\mathcal{L}_{MMD-IR} \quad (5)$$

where $\mathcal{L}_{CE}$ denotes source supervised loss using source labels, and $\mathcal{L}_{Con}$ denotes the geometric domain alignment loss based on the mutual nearest neighbor contrastive learning, and $\mathcal{L}_{MMD-IR}$ denotes the statistical domain alignment loss using instance re-weighting scheme. $\lambda$ and $\mu$ are two trade-off parameters.

## 4 Experiments

In this section, we conduct extensive experiments to evaluate the effectiveness of the proposed framework (MNCRI).

### 4.1 Experimental Settings

#### 4.1.1 Dataset

In our work, we conduct experiments on three open-source software project datasets: FFmpeg, QEMU (Zhou et al., 2019) and ReVe (Chakraborty et al., 2021). There are several reasons for choosing the above datasets as experimental datasets. The first one is that they are manually labeled so that they could provide accurate supervised information.

Second, the number of samples of these datasets is abundant to support model training. The implementation details can be found in Appendix A.

### 4.1.2 Baselines

The state-of-the-art cross-domain code vulnerability detection methods include: VulDeePecker (Li et al., 2018), SySeVr (Li et al., 2021), Devign (Zhou et al., 2019), CodeBERT (Feng et al., 2020), GraphCodeBERT (Guo et al., 2020), CodeT5 (Wang et al., 2021), SCDAN (Nguyen et al., 2019), CD-VulD (Liu et al., 2020), and VulGDA (Li et al., 2023).

### 4.1.3 Evaluation Metric

In general, F1-score is a widely used metric to evaluate the performance of vulnerability detection systems. Let $TP$ be the number of vulnerable samples that are predicted correctly, $FP$ be the number of truly non-vulnerable samples that are predicted as vulnerable and $FN$ be the number of truly vulnerable samples that are predicted as non-vulnerable. Precision ($P = \frac{TP}{TP+FP}$) represents the ratio of vulnerable samples that are correctly predicted as vulnerable out of all samples predicted as vulnerable. Recall ($R = \frac{TP}{TP+FN}$) represents the ratio of vulnerable samples that are correctly predicted as vulnerable out of all truly vulnerable samples. The F1-score ($F1 = \frac{2 \times P \times R}{P+R}$) is the harmonic mean of precision and recall. It is clear that the higher F1-score represents better classification performance. Besides, vulnerability detection tasks also focus on TPR (True Positive Rate) performance (as known as Recall) of detectors. Because the higher TPR represents that the more vulnerable samples are discriminated correctly, it is crucial for the performance evaluation of detectors. Therefore, in our experiments, we adopt TPR (True Positive Rate) and F1-score as our evaluation metrics.

### 4.2 Results

We conduct extensive experiments to verify the effectiveness of the proposed framework (MN-CRI). Experimental results are shown in Table 1. Our proposed framework MNCRI (CodeT5) achieves the best performances on all cross-domain code vulnerability detection tasks, and outperforms all baseline methods. For instance, in the FFmpeg-to-QEMU (F → Q) cross-domain detection task, MNCRI (CodeT5) achieves an F1-score of 58.21%, outperforming the current state-of-the-art cross-domain detection method VulGDA by

5.41%. Meanwhile, MNCRI (CodeT5) obtains a TPR gain of 9.18% compared to VulGDA. Moreover, our framework equipped with CodeBERT, GCBERT (GraphCodeBERT) also can obtain significant performance improvements compared to baseline methods. This demonstrates that the proposed framework has good generalization performance and it does not rely on the specific code feature extractor and can be adapted to various feature extractors. Since CodeT5 adopts more effective pre-training mechanisms compared to Code-BERT and GCBERT (GraphCodeBERT), MNCRI using CodeT5 as the feature extractor obtains the largest performance improvements. This demonstrates that a power feature extractor is helpful for improving cross-domain vulnerability detection performance. To make the comparison with Vul-GDA, MNCRI (VulGDA-f) uses the same feature extractor as VulGDA. We can observe that MNCRI (VulGDA-f) consistently outperforms VulGDA. We can conclude that the performance improvements can be attributed to the proposed domain adaptation strategies.

By analyzing experimental results, we can observe that MNCRI could significantly improve TPR (**T**rue **P**ositive **R**ate) of the detector compared to other cross-domain vulnerability detection methods. This observation shows that mutual nearest neighbor contrastive learning and the instance reweighting scheme are conducive to reducing domain discrepancy by jointly conducting geometric domain alignment and statistical domain alignment. Besides, compared to the detectors CD-VulD, Vul-GDA, and MNCRI exploiting domain adaptation techniques, traditional detectors VulDeePecker, SySeVr and Devign have the worse performance in cross-domain detection tasks. This is because traditional detectors do not consider the domain gap between domains. The comparison results indicate the importance of domain adaptation in cross-domain code vulnerability detection tasks.

In order to more intuitively demonstrate the classification performance of MNCRI in cross-domain vulnerability detection tasks, we plot the histogram distributions of softmax-based prediction confidence on the target domain dataset (see Figure 2). We can see that the prediction confidence scores of VulGDA are mainly distributed around 0.5 (see Figure 2(a)). This phenomenon indicates that VulGDA has no discriminative ability for a large number of target instances and is unable to judge whether

| Methods | F→Q | | F→R | | Q→F | | Q→R | | R→F | | R→Q | |
|---|---|---|---|---|---|---|---|---|---|---|---|---|
| | TPR | F1 | TPR | F1 | TPR | F1 | TPR | F1 | TPR | F1 | TPR | F1 |
| VulDeePecker | 64.82 | 41.31 | 12.59 | 12.47 | 34.85 | 32.04 | 33.98 | 12.04 | 35.57 | 29.01 | 16.57 | 13.88 |
| SySeVr | 66.98 | 43.22 | 12.73 | 13.19 | 35.57 | 33.22 | 35.77 | 12.96 | 37.28 | 30.05 | 17.65 | 14.36 |
| Devign | 68.92 | 45.61 | 14.58 | 13.31 | 37.99 | 35.91 | 39.82 | 13.55 | 35.56 | 31.35 | 18.97 | 16.28 |
| CodeBert | 53.22 | 46.66 | 10.66 | 11.22 | 37.44 | 38.15 | 30.02 | 12.97 | 30.66 | 33.59 | 15.21 | 18.34 |
| GCBERT | 55.34 | 47.14 | 10.79 | 11.30 | 39.40 | 38.60 | 31.09 | 13.68 | 30.97 | 33.98 | 16.88 | 18.77 |
| CodeT5 | 58.57 | 49.50 | 11.23 | 12.11 | 38.80 | 39.42 | 33.70 | 15.02 | 31.77 | 35.73 | 18.22 | 20.39 |
| SCDAN | 70.30 | 52.32 | 13.99 | 12.68 | 40.78 | 43.30 | 43.29 | 17.65 | 38.92 | 37.05 | 15.29 | 22.21 |
| CD-VulD | 73.21 | 51.24 | 13.88 | 14.06 | 48.80 | 46.68 | 48.80 | 20.10 | 40.25 | 40.17 | 18.71 | 25.78 |
| VulGDA | 76.76 | 52.80 | 15.22 | 14.37 | 47.96 | 47.72 | 50.52 | 20.98 | 40.68 | 42.86 | 19.88 | 26.35 |
| MNCRI (VulGDA-f) | 81.37 | 54.95 | 16.21 | 15.44 | 50.39 | 50.99 | 51.39 | 22.37 | 41.38 | 45.92 | 21.69 | 29.32 |
| MNCRI (CodeBERT) | 84.55 | 56.57 | 17.11 | 16.68 | 53.22 | 53.96 | 53.88 | 23.10 | 42.56 | 48.10 | 21.69 | 29.32 |
| MNCRI (GCBERT) | 84.92 | 57.09 | 17.92 | 17.02 | 54.10 | 54.11 | 54.74 | 24.97 | 44.74 | 49.83 | 22.28 | 30.10 |
| MNCRI (CodeT5) | **85.94** | **58.21** | **18.66** | **17.51** | **56.70** | **55.42** | **57.09** | **25.65** | **46.60** | **50.22** | **24.41** | **31.50** |

Table 1: Results on cross-domain code vulnerability detection tasks. For simplicity, GCBERT denotes the Graph-CodeBERT (Guo et al., 2020) model. MNCRI (VulGDA-f) denote that MNCRI use the same feature extractor as VulGDA. MNCRI (CodeT5) denotes that our framework MNCRI uses CodeT5 as the feature extractor. F denotes the FFmpeg project dataset, and Q denotes the QEMU project dataset, and R denotes the Reveal project dataset. F → Q represents the source domain is the FFmpeg project dataset and the target domain is the QEMU dataset.

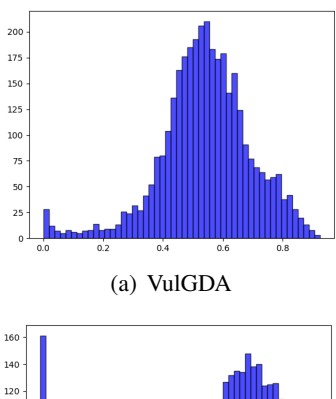

(a) VulGDA

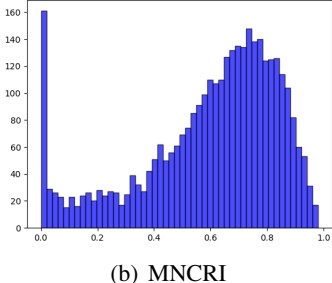

(b) MNCRI

Figure 2: Histogram of softmax-based prediction confidence on the target domain test dataset. (a) is the prediction confidence score of VulGDA; (b) is the prediction confidence score of MNCRI. x-axis coordinate represents the prediction confidence score from 0 to 1 and y-axis coordinate represents the corresponding counts.

target instances are vulnerable or non-vulnerable. In contrast, our proposed framework MNCRI overcomes this phenomenon (see Figure 2(b)). We can observe that there are few prediction confidences distributed around 0.5, which shows that MNCRI learns the more accurate vulnerable patterns across domains and has the superior discriminative ability for target instances.

Overall, the promising performance of MNCRI can be attributed to the mutual nearest neighbor-based contrastive learning and the instance re-weighting scheme.

### 4.3 Ablation Study

To analyze the impact of different components in our proposed MNCRI on the performance, we conduct an ablation study on all cross-domain vulnerability detection tasks and report the results in Table 2.

**Effect of Mutual Nearest Neighbors Contrastive Learning.** From the results reported in Table 2, we can observe that the removal of the mutual nearest neighbors contrastive learning ("**w/o** $\mathcal{L}_{Con}$") sharply reduces the performance in all evaluation metrics (TPR and F1). This indicates that contrastive learning based on mutual nearest neighbors guarantees geometric domain alignment by pulling mutual nearest neighbors pairs closer to each other and pulling away non-geometrically close samples. Furthermore, we remove the memory bank mechanism ("**w/o MB**") in the contrastive learning training stage and record its results. We can find that removing the memory bank mechanism would degrade performance. This indicates that the global semantic information containing in the memory bank is conducive to achieving more accurate domain alignment.

**Effect of Instance Re-weighting Scheme.** To evaluate the contribution of the instance re-weighting scheme, we abandon it and report its results in Table 2. We can see that the removal of the instance re-weighting scheme ("**w/o IR**") leads to performance degradation. This implies that assigning different weights to different instances according to relevance is helpful for reducing distributional divergence between domains, which en-

| Methods | F → Q | | F → R | | Q → F | | Q → R | | R → F | | R → Q | |
|---|---|---|---|---|---|---|---|---|---|---|---|---|
| | TPR | F1 | TPR | F1 | TPR | F1 | TPR | F1 | TPR | F1 | TPR | F1 |
| MNCRI (Full) | **85.94** | **58.21** | **18.66** | **17.51** | **56.7** | **55.42** | **57.09** | **25.65** | **46.60** | **50.22** | **24.41** | **31.50** |
| MNCRI (w/o $\mathcal{L}_{Con}$) | 79.74 | 54.30 | 16.92 | 15.29 | 49.99 | 49.37 | 52.69 | 22.13 | 42.72 | 45.35 | 20.61 | 28.17 |
| MNCRI (w/o MB) | 83.49 | 55.76 | 17.24 | 16.66 | 53.85 | 52.28 | 55.37 | 23.48 | 43.92 | 48.50 | 23.69 | 30.22 |
| MNCRI (w/o IR) | 84.87 | 56.87 | 18.04 | 16.65 | 55.12 | 54.39 | 56.21 | 24.71 | 45.98 | 49.74 | 24.01 | 30.85 |

Table 2: Experimental results of ablation study.

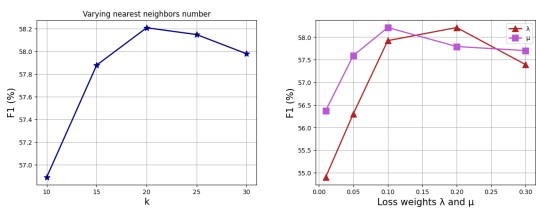

(a) Neighbors number $k$     (b) Loss weights $\lambda$ and $\mu$

Figure 3: (a) Varying nearest neighbors number $k$. (b) Varying loss weights $\lambda$ and $\mu$.

ables the model to derive better cross-domain representations, and thus leads to improved MNCRI performance.

### 4.4 Sensitivity to hyper-parameter

**Impact of the nearest neighbors number $k$.** To show the sensitivity of MNCRI to the nearest neighbors number $k$, we conduct the experiments on the F → Q cross-domain vulnerability detection task, and show the results in Figure 3(a). We observe that the F1-score increases with the increasing value of $k$ and peaks at k = 20. Further increasing the values of $k$ results in worse performance. The F1-score varies slightly within a wide range of $k$, demonstrating that MNCRI is robust to the choices of $k$.

**Impact of loss weights $\lambda$ and $\mu$.** To show the sensitivity of MNCRI to the loss weights $\lambda$ and $\mu$, we conduct control experiments on the F → Q cross-domain vulnerability detection task. The results are illustrated in Figure 3(b). We can observe that MNCRI achieves the best performance when $\lambda$ is set to 0.2. Within a wide range of $\lambda$, the performance of MNCRI changes very little, validating that MNCRI is stable to the choices of $\lambda$. Similarly, MNCRI varies slightly with $\mu \in [0.01, 0.3]$, showing that MNCRI is robust to the selection of $\mu$.

### 4.5 Analysis of Few-Shot Condition

To evaluate the generalizability of MNCRI in few-shot vulnerability detection, we also evaluate MN-CRI in the few-shot condition on the F → Q cross-

domain vulnerability detection task. From the experimental results shown in Table 3, we can see that MNCRI outperforms all baseline methods under the few-shot condition. This verifies the effectiveness and generalizability of MNCRI in dealing with both zero-shot and few-shot vulnerability detection.

| Methods | TPR | f1 |
|---|---|---|
| SCDAN | 74.51 | 52.67 |
| CD-VulD | 79.32 | 53.45 |
| VulGDA | 79.05 | 54.88 |
| MNCRI | 86.33 | 59.89 |

Table 3: Experimental results of few-shot condition with 20% target domain labeled samples for training.

### 4.6 Conclusion

In this paper, we propose a novel cross-domain code vulnerability detection framework named MNCRI to achieve domain alignment. On the one hand, we introduce mutual nearest neighbors contrastive learning to learn geometric relationships between source domain samples and target domain samples, so as to achieve geometric domain alignment. On the other hand, we introduce the instance re-weighting scheme to reduce distributional divergence between domains and achieve statistical domain alignment. Finally, we conduct extensive experiments to verify the effectiveness of our proposed framework.

### Limitations

Here we summarize the limitations for further discussion and investigation of the community. Our proposed $k$-nearest neighbor mechanism requires manually setting the optimal $k^*$ to achieve the best cross-domain detection performance, which is inefficient. A better solution is to design an adaptive $k$-nearest neighbor mechanism which could adaptively seek optimal neighbor samples according to the semantic information of training samples.

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

## A   Implementation Details

We use CodeBERT, GraphCodeBERT, CodeT5 as the feature extractor, respectively. We use cross-entropy to calculate the classification loss. The classifier is a 2-layer full connected layers with 256-dimensional output, and uses MMD to measure the distributional domain discrepancy. The initial learning rate is set to 5e-4. We split source and target samples into different mini-batches with size 36. The total training epoch is 10. The temperature parameter $\tau$ is set to 0.05.