# OpenReview forum: "Joint Geometrical and Statistical Domain Adaptation for Cross-domain Code Vulnerability Detection"
_EMNLP/2023/Conference — EMNLP 2023 Main_

### Official Review · Reviewer_HWdy · 2023-08-02

**Typos Grammar Style And Presentation Improvements:** N/A
**Soundness:** 4

**Excitement:**

3: Ambivalent: It has merits (e.g., it reports state-of-the-art results, the idea is nice), but there are key weaknesses (e.g., it describes incremental work), and it can significantly benefit from another round of revision. However, I won't object to accepting it if my co-reviewers champion it.

**Missing References:**

N/A

**Paper Topic And Main Contributions:**

This paper studies a very interesting problem, i.e., cross-domain code vulnerability detection. The authors argued that prior work ignores the negative effect of private semantic characteristics of the target domain for alignment, and proposed a new approach based on mutual nearest neighbor contrastive learning. Furthermore, the authors introduced an instance re-weighting scheme to enhance the alignment between two domains. Experiments on three open-source software project datasets demonstrate the effectiveness of the proposed method.

**Questions For The Authors:**

1. Please clarify the novelty of the proposed method, especially the contrastive learning module and the instance re-weighting module.
2. Please discuss the sensitivity of the hyperparameters.

**Reasons To Accept:**

1. Overall, this paper is clearly organized and well written. The studied problem of cross-domain code vulnerability detection is very interesting.
2. The authors revealed limitations in existing work, and further motivated the MNCRI approach.
3. The proposed method can help address the negative transfer problem, which is a long-standing problem in transfer learning.
4. Experimental results show that the proposed method performs much better than baselines. Also, ablation studies verified the contributions of each component of the proposed model.

**Reasons To Reject:**

1. The novelty of proposed method is not significant. In particular, the proposed model combines contrastive learning and MMD. The instance re-weighting scheme is only a minor modification of the well known MMD approach.
2. The proposed model has two hyperparameters, \lambda and \mu. Figure 3(b) shows that the F1 score is quite sensitive to minor changes of hyperparameters, considering the small range of [0.01, 0.3].
3. It seems the proposed MNCRI is a general approach, and the authors may discuss its potential applications in other domains.

**Reproducibility:**

4: Could mostly reproduce the results, but there may be some variation because of sample variance or minor variations in their interpretation of the protocol or method.

**Reviewer Confidence:**

3: Pretty sure, but there's a chance I missed something. Although I have a good feel for this area in general, I did not carefully check the paper's details, e.g., the math, experimental design, or novelty.

---

> ### Author Rebuttal · Authors · 2023-08-28
>
> # The novelty of the proposed method
> In this paper, we propose a cross-domain code vulnerability detection framework, which jointly achieves geometric domain alignment and statistical domain alignment by introducing
> the contrastive learning paradigm and the instance re-weighting scheme.
>
> **The novelty of the contrastive learning module.**  In cross-domain code vulnerability detection tasks, different domains may have their own private semantic characteristics. Previous studies forcibly align the source domain and target domain and do not consider the negative effect of private semantic characteristics for domain alignment, which easily causes the problem of negative transfer. To address this problem, we propose a mutual nearest neighbor contrastive learning paradigm to achieve geometric domain alignment. Specifically, we first propose a novel geometric domain alignment principle: semantically consistent instances should be geometrically adjacent to each other, whether within or across domains. Based on this principle, we propose the mutual nearest neighbor contrastive learning paradigm, which pulls instances with the consistent semantic across the source and target domains closer to each other and pushes away those semantically inconsistent instances. In this way, it achieves the alignment of common semantic characteristics across domains while the private semantic characteristics of each domain are separated, which solves the problem of negative transfer. Compared to previous methods, our proposed mutual nearest neighbor contrastive learning paradigm achieves geometric domain alignment by exploring the geometric structure relationship of source and target domains in feature space.
>
> **The novelty of the instance re-weighting module.**  The re-weighting scheme is a general method that is widely used in various deep learning application scenarios, such as attention mechanisms equipped in Transformer and instance re-weighting methods for imbalanced classification. Different domains adopt different instance re-weight schemes to address their own domain issues according different characteristics of domains.  These proposed instance re-weighting schemes bring performance improvements and have been proven to be novel. In our work, we mainly focus on improving cross-domain code vulnerability detection performance. In cross-domain code vulnerability detection tasks, some target instances that are not relevant to source instances. Previous studies reduce the domain discrepancy by forcibly minimizing statistical distribution divergence (MMD) of the entire source domain and the target domain, which easily leads to the problem of excessive alignment. In our work, we argue that assigning large weights to relevant instances and small weights to irrelevant instances could alleviate the problem of excessive alignment and improve cross-domain vulnerability detection performance. Compared to previous instance re-weighting research works that utilize solvers or complex optimization methods to obtain instance weights, our proposed instance re-weighting scheme is simple yet effective. Specifically, we adopt the prediction confidence of the classifier trained on the source domain for target instances as the weights of target instances during statistical domain alignment. Introducing the instance re-weighting scheme mitigates the adverse influence of some target instances that are not relevant to source instances, which is helpful for alleviating the problem of excessive alignment. Experimental results prove that the effectiveness of the proposed instance re-weighting scheme.
>
>
>
>
> # The sensitivity of the hyperparameters
> As shown in Figure 3(b), our method achieves the stable detection performance when $\mu \in
> [0.1,0.3]$. When $\mu$ is set to 0.01, the performance drops. But it cannot conclude that the F1 score is sensitive to minor changes of hyperparameters, as 0.01 is ten times smaller than 0.1. The performance degradation caused by differences in order of magnitude cannot be avoided in many machine learning application scenarios. In our work, the total objective function is formulated as:
>
> $L = L_{CE} + L_{Con} + L_{MMD-IR}$
>
> where hyperparameter $\mu$ represents the weights of the instance re-weighting module in the overall objective function optimization. If $\mu$ is set to a relatively small value (e.g., $\mu=0.01$), it indicates that the instance re-weighting module ($L_{MMD-IR}$) will play a relatively small role on the overall objective optimization.  For instance, if the detection performance does not significantly decrease when $\mu$ is set to a relatively small value (e.g., $\mu=0.01$), it can be concluded that the instance re-weighting module is useless and performance improvements come from other modules rather than the instance re-weighting module. Because the instance re-weighting module is close to being banned. Therefore, as illustrated in Figure3(b), when $\mu=0.01$, the performance degradation is due to the instance re-weighting module being disabled. On the contrary, when $\mu$ is set to a large value (e.g., $\mu \in [0.1, 0.3]$), our method achieves stable detection performance, showing that our proposed method is robust to the selection of $\mu$.
>
>
> # About potential applications
> We propose a novel cross-domain code vulnerability detection framework named MNCRI, which aims to align the common semantic characteristics between domains and separate out private semantic characteristics of each domain. Certainly, our method is not only applicable to cross-domain code vulnerability detection tasks. It can be applied to some knowledge-transfer application scenarios, such as cross-lingual sentiment classification. Some languages derive from the same language family, and they have some common ways of expressing emotions. Simultaneously, due to differences in regions, these languages also have their own unique ways of expressing emotions. Our method MNCRI can be employed to transfer common emotional expression patterns between different languages while separating their respective unique emotional expression patterns, so as to improve the performance of cross-lingual sentiment classification. Furthermore, our method can be applied to some cross-domain news classification tasks where different types of news share common semantic characteristics while also having their own private semantic characteristics.
>
> In this paper, we mainly focus on improving the performance of cross-domain code vulnerability detection.  Potential applications are not discussed in our original paper. Thanks for your insightful suggestions. We will depict the above potential applications of MNCRI in the revised paper.
>
>
>
> Thanks for your insightful suggestions. We will polish the paper based on these valuable points in the revised version.

---

### Official Review · Reviewer_f7o6 · 2023-08-04

**Typos Grammar Style And Presentation Improvements:** No
**Soundness:** 4

**Excitement:**

3: Ambivalent: It has merits (e.g., it reports state-of-the-art results, the idea is nice), but there are key weaknesses (e.g., it describes incremental work), and it can significantly benefit from another round of revision. However, I won't object to accepting it if my co-reviewers champion it.

**Missing References:**

No

**Paper Topic And Main Contributions:**

This paper addresses the issue of cross-domain code vulnerability detection, where a detector trained on a source domain fails to provide accurate predictions on unseen target domains. The main contributions of this paper include:
1) The proposal of a novel cross-domain code vulnerability detection framework, MNCRI.
2) The introduction of a mutual nearest neighbor contrastive learning paradigm to align the source and target domains geometrically and separate the private semantic characteristics of each domain.
3) The incorporation of an instance re-weighting scheme to mitigate excessive alignment and improve domain alignment in statistical domain.

**Questions For The Authors:**

1. The details of the datasets provided in the appendix appear to have a balanced distribution of positive and negative examples. However, in real-world applications, scenarios with fewer vulnerabilities may introduce additional difficulties.
2. If the code has many different types of vulnerabilities with significant characteristic differences, considering all diverse vulnerability features as a single category (i.e., “vulnerable”) in a binary classification model may be problematic.
3. Can the authors provide an ablation experiment that eliminates the use of Maximum Mean Discrepancy (MMD) loss for statistical domain alignment to understand whether statistical domain alignment without the Instance Re-weighting Scheme has a negative or small positive effect on experimental results?

**Reasons To Accept:**

1. This paper addresses a significant issue in the field of code vulnerability detection, where models trained on source domains often fail to generalize to target domains. This is a relevant and practical problem of interest to the NLP community.
2. The proposed framework, MNCRI, introduces a novel approach to cross-domain code vulnerability detection by combining mutual nearest neighbor contrastive learning and instance re-weighting, showing promising results in improving domain alignment and performance.
3. The paper provides extensive experimental evaluations that demonstrate the effectiveness of the proposed framework, with results showing significant performance improvements over baseline methods and highlighting the contributions made by this work.

**Reasons To Reject:**

1. The paper lacks detailed descriptions of the implementation details of nearest neighbor instance selection, such as the feature used for calculation and the distance metric employed. Additionally, it is unclear whether the affinity relationship matrix is initialized only once during the training process or changes as training progresses.
2. The paper also lacks clarity regarding the details of the memory bank, including how many samples’ features are stored.
3. The assumption that similar instances have common semantic characteristics while dissimilar instances have private semantic characteristics may be overly idealistic, as dissimilar instances may also share common semantic characteristics, such as different manifestations of the same vulnerability across different software projects.

**Reproducibility:**

4: Could mostly reproduce the results, but there may be some variation because of sample variance or minor variations in their interpretation of the protocol or method.

**Reviewer Confidence:**

4: Quite sure. I tried to check the important points carefully. It's unlikely, though conceivable, that I missed something that should affect my ratings.

---

> ### Author Rebuttal · Authors · 2023-08-28
>
> # About vulnerability dataset
> In our work, in order to make a fair comparison with other baseline methods, we adopt the same datasets as baselines. The current published vulnerability datasets commonly have a balanced ratio of positive and negative examples to learn discriminative vulnerability representations. As you described, in real-world applications, the number of vulnerable samples is less than that of non-vulnerable samples. The class imbalance problem is a hot research topic in deep learning field. Some research works was proposed to solve class imbalance problem, such as focal loss, re-sampling strategies, etc. Due to the lack of imbalanced datasets in the code vulnerability detection field, we cannot report corresponding performance. But our method can solve class imbalance problem by equipping with class-imbalance learning methods (e.g., focal loss or re-sampling strategies).
>
> # About binary classification model
> The goal of the code vulnerability detection task is to determine whether a piece of code is secure and non-vulnerable rather than to identifying types of vulnerabilities exist in the code. Following previous research works, such as VulGDA, vulnerability detection is viewed as a binary classification problem. Of course, different types of vulnerabilities may have different characteristics. But the semantic characteristics of non-vulnerable code samples are consistent and are distinctly different from the semantic characteristics of vulnerable samples. Therefore, current research works and vulnerability dataset collection works are mainly based on binary classification tasks. Furthermore, published vulnerability datasets with multiple types of vulnerabilities are scarce due to the expensive annotation costs. We appreciate your insightful suggestions. We plan to build multi-category datasets and explore multi-class code vulnerability detection tasks in our future work.
>
> # About the ablation experiment that eliminates the use of Maximum Mean Discrepancy (MMD) loss for statistical domain alignment
>
> We conducted an ablation experiment that eliminates the use of Maximum Mean Discrepancy (MMD) loss for statistical domain alignment. The results are shown in the following table. MNCRI (w/o L_MMD) denotes that our proposed method without the MMD loss for statistical domain alignment. We can see that omitting the MMD loss for statistical domain alignment results in a decrease in performance. This demonstrates that statistical domain alignment without the Instance Re-weighting Scheme still has a positive effect on experimental results and plays a constructive role in the cross-domain code vulnerability detection tasks.
> |Methods|F -> Q|F -> R|Q -> F|Q -> R|R -> F|R -> Q|
> |:------------------------------------|:---:|:---:|:---:|:---:|:---:|:---:|
> |MNCRI (Full) |58.21|17.51|55.42|25.65|50.22|31.50|
> |MNCRI (w/o L_Con) |54.30|15.29|49.37|22.13|45.35|28.17
> |MNCRI (w/o IR)    |56.87 |16.65|54.39|24.71|49.74|30.85|
> |MNCRI (w/o L_MMD)   |55.75|16.04|53.47|22.96 |47.98 |29.92|
>
> # About implementation details of nearest neighbor instance selection
> As shown in Figure.1, we first use a feature extractor (e.g., CodeT5) to extract semantic features of samples and the feature dimension is 768. In other words, the output of the last layer of the feature extractor serves as the semantic features of samples. Next, we employ the Cosine distance as the distance metric to measure the distance between sample semantic features. For a sample $x_i^t$ in the target domain, we retrieve its $k$ nearest neighbors $H_i^s$ in the source domain. Similarly, for a sample $x_j^s$ in the source domain, we retrieve its $k$ nearest neighbors $H_j^t$ in the target domain. Subsequently, the affinity relationship matrix $M^{st}$ is built by following the rule: $M_{i,j}^{st}$= 1 if only $x_i^t$ and $s_j^s$ are the mutual nearest neighbor pair; otherwise $M_{i,j}^{st}$= 0. Finally, we calculate the mutual nearest neighbor contrastive loss based on the built affinity relationship matrix  (refer to Equation.2 ) and optimizing model weights. We repeat the above steps in each training iteration step until the model converges. Thus, in our work, the affinity relationship matrix is updated (changed) in each training iteration step.  We will provide the missing implementation details in the revised paper.
>
> # About implementation details of the memory bank
> In our experiments, the memory bank stores the previous five batches of samples’ features. The number of stored samples’ features is $5 \times 36 \times 2 = 360$ (the batch size is 36, including source domain samples’ features and target domain samples’ features). The memory bank is updated using a queue mechanism.
>
> # About the assumption that similar instances have common semantic characteristics
> In our work, we argue the instances with common semantic characteristics have
> similar semantic features rather than these instances themselves being similar. As you mentioned, the same types of vulnerabilities may have different manifestations in different projects. But they still have common vulnerable patterns (common semantic characteristics). For instance, CWE-119 (Buffer Errors) vulnerability functions have different manifestations in different projects due to program style and application scenarios. But their vulnerable patterns are the same and consistent. That is, all vulnerability functions trigger data buffer overflow errors. The goal of the detection model is to extract common vulnerable patterns (common semantic characteristics) from different manifestations. Therefore, the semantic features of instances with common semantic characteristics are similar in the feature space.
>
> Thanks for your insightful suggestions. We will polish the paper based on these valuable points in the revised version.

---

### Official Review · Reviewer_hFha · 2023-08-11

**Soundness:** 4

**Excitement:**

3: Ambivalent: It has merits (e.g., it reports state-of-the-art results, the idea is nice), but there are key weaknesses (e.g., it describes incremental work), and it can significantly benefit from another round of revision. However, I won't object to accepting it if my co-reviewers champion it.

**Paper Topic And Main Contributions:**

Centering on the objective of detecting code vulnerabilities across different domains, the paper tackles the challenges of negative transfer and excessive alignment. The primary contributions of this study encompass the introduction of an innovative framework, MNCRI, which leverages mutual nearest neighbor contrastive learning and instance reweighing to achieve enhanced alignment. Through extensive experiments, the approach is shown to markedly outperform existing state-of-the-art methods for cross-domain code vulnerability detection.

**Reasons To Accept:**

- The identification of the two proposed problems, namely negative transfer and excessive alignment, provides an intriguing perspective.
- The performance achieved in the experiments is surprisingly impressive.

**Reasons To Reject:**

- There appears to be some confusion regarding the fairness of the comparison in the results. The baseline methods, such as SCDAN, CD-VulD, and VulGDA, referenced in the paper, were not evaluated under the same experimental conditions (like dataset). This has led to uncertainty about whether the competitive results presented in the paper were reproduced by the authors themselves. While the supplementary code is provided, it lacks inclusion of the above referred method. Additionally, considering that SCDAN is solely LSTM-based, it might be beneficial to incorporate PLM-based techniques along with traditional domain adaptation approaches (e.g., DANN). This would help emphasize the significance of the proposed problem—negative transfer and excessive alignment—and consequently validate the effectiveness of the proposed methodology.

- The level of novelty attributed to the proposed method for cross-domain code vulnerability detection appears to be somewhat limited. It's worth noting that k-Nearest Neighbor based adaptation [1] has previously been explored, and the concept of instance weighting [2] is not a new or groundbreaking technique. But admittedly, this paper is the first to apply these methods within the specific field of cross-domain code vulnerability detection.
[1] Sener, Ozan, Hyun Oh Song, Ashutosh Saxena, and Silvio Savarese. "Learning Transferrable Representations for Unsupervised Domain Adaptation." NIPS (2016).
[2] Sun, Qian, Rita Chattopadhyay, Sethuraman Panchanathan, and Jieping Ye. "A Two-Stage Weighting Framework for Multi-Source Domain Adaptation." NIPS (2011).


**Reproducibility:**

4: Could mostly reproduce the results, but there may be some variation because of sample variance or minor variations in their interpretation of the protocol or method.

**Reviewer Confidence:**

4: Quite sure. I tried to check the important points carefully. It's unlikely, though conceivable, that I missed something that should affect my ratings.

**Typos Grammar Style And Presentation Improvements:**

- It seems that there is a missing illustration for Equation 3 in the context you provided. IIUC, maybe U should be written in n_s, and W should be written in n_t.

- The organization of the introduction appears to be problematic, as it attempts to convey a significant amount of information within just two paragraphs spanning two pages. This dense presentation makes it challenging for readers to grasp the content effectively.

---

> ### Author Rebuttal · Authors · 2023-08-28
>
> # About baseline methods
> In this paper, to make a fair comparison, we adopt the same experimental conditions (including datasets and experimental settings) as the latest baseline VulGDA. Since links are prohibited in the rebuttal material, we will release the source code of baselines in the revised paper. To verify our proposed method does not rely on specific feature extractors (e.g., PLM models), we conduct supplementary experiments that adopt the same model architecture as SCDAN without using PLM models. The results are shown in the following table. Compared to SCDAN, MNCRI obtains an average relative F1 gain of 3.1%. This indicates that our method does not rely on the specific feature extractor.
>
> |Methods|F -> Q|F -> R|Q -> F|Q -> R|R -> F|R -> Q|
> |:------------------------------------|:---:|:---:|:---:|:---:|:---:|:---:|
> | SCDAN |52.32|16.66|52.28|23.48|48.50|30.22|
> | MNCRI(SCDAN-f) |56.87|16.65|54.39|24.71|49.74|30.85|
>
> # About novelty attributed to the proposed method
> Although our proposed method and the approach presented in [1] both employ the k-nearest neighbor mechanism, our method is fundamentally inconsistent with paper [1]. Paper [1] infers target labels by integrating the labels of $k$ neighbors of target samples from the source domain. Our method first retrieves $k$ nearest neighbors of source samples from the target domain and $k$ nearest neighbors of target samples from the source domain. Subsequently, we construct source-target mutual nearest neighbor pairs by follow the rule: if a source sample $x_i^s$ is the neighbor of a target sample $x_j^t$ and $x_j^t$ is the neighbor of $x_i^s$, $x_i^s$ and $x_j^t$ are the mutual nearest neighbor pair. Finally, we exploit a novel contrastive learning paradigm that pulls these nearest neighbors closer to each other and pushing away those non-nearest neighbor samples. In this way, it enables the model to learn more discriminative representations and achieve better semantic transfer. Compared to our method, paper [1] only uses the k-nearest neighbor mechanism to predict target labels and does not focus on learning discriminative cross-domain representations.
>
> The re-weighting scheme is a general method that is widely used in various deep learning application scenarios, such as attention mechanisms equipped in Transformer and instance re-weighting methods for imbalanced classification. Different domains adopt different instance re-weight schemes to address their own domain issues according the different characteristics of domains.  These proposed instance re-weighting schemes bring performance improvements and have been proven to be novel. In our work, we mainly focus on improving cross-domain code vulnerability detection performance. In cross-domain vulnerability detection tasks, there are some target instances that are not relevant to source instances. Previous studies reduce the domain discrepancy by forcibly minimizing statistical distribution divergence (MMD) of the entire source domain and the target domain, which easily leads to the problem of excessive alignment. In our work, we argue that assigning large weights to relevant instances and small weights to irrelevant instances could alleviate the problem of excessive alignment and improve cross-domain code vulnerability detection performance. Compared to previous instance re-weighting research works that utilize solvers (e.g., such as the research work presented in [2]) or complex optimization methods to obtain instance weights, our proposed instance re-weighting scheme is simple yet effective. Specifically, we adopt the prediction confidence of the classifier trained on the source domain for target instances as the weights of target instances during statistical domain alignment. Introducing the instance re-weighting scheme mitigates the adverse influence of some target instances that are not relevant to source instances, which is helpful for alleviating the problem of excessive alignment. Experimental results prove that the effectiveness of the proposed instance re-weighting scheme.
>
> [1] Sener, Ozan, Hyun Oh Song, Ashutosh Saxena, and Silvio Savarese. "Learning Transferrable Representations for Unsupervised Domain Adaptation." NIPS (2016).
>
> [2] Dynamic re-weighting for long-tailed semi-supervised learning.  NIPS (2011).
>
>
>
> # About Equation 3
> We apologize for our imprecise formula definition. In Equation 3, $U = n_s$ and $W = n_t$. Thanks for your valuable suggestions. We will fix this problem in our revised paper.
>
> Thanks for your insightful suggestions. We will polish the paper based on these valuable points in the revised version.

---

### Meta-Review · Area_Chair_c9Zx · 2023-09-19

**Recommendation:** 4

**Metareview:**

The reviewers were enthusiastic about the paper and agreed that the paper has significant technical merits that were well-justified and supported well by the experiments in the paper. The rebuttal process helped clarify the technical contributions of the work. While the new results will need to be included before the paper is published, there is strong interest in this line of work and the excitement on the proposed approaches is high.

---

### Decision · Program_Chairs · 2023-10-07

**Decision:**

Accept-Main

**Comment:**

The reviewers were enthusiastic about the paper and agreed that the paper has significant technical merits that were well-justified and supported well by the experiments in the paper. The rebuttal process helped clarify the technical contributions of the work. While the new results will need to be included before the paper is published, there is strong interest in this line of work and the excitement on the proposed approaches is high.